# Effects of High-Temperature Tempering on Mechanical Properties and Microstructure of SA738 Gr.B Steel

**Yanmei Li [1,\*], Shuzhan Zhang [1,\*], Chunyao Zhao [1], Minghui Song [1] and Zaiwei Jiang [2]**

[1] The State Key Laboratory of Rolling and Automation, Northeastern University, Shenyang 110819, China; zhaochunyao14@foxmail.com (C.Z.); songminghui19@foxmail.com (M.S.)

[2] Nanjing Iron and Steel Co., Ltd., Nanjing 210035, China; jiangzaiwei@njsteel.com.cn

\* Correspondence: liym@ral.neu.edu.cn (Y.L.); shuzhan-zhang@foxmail.com (S.Z.)

**Abstract:** In this paper, JMatPro thermodynamic software, OM, SEM, TEM, and EPMA were used to study the microstructure and mechanical properties of SA738 Gr.B nuclear power steel after tempering at 630–710 °C. When tempered within the range of 630–670 °C, a huge amount of $M_3C$ and MC carbides were dispersed and precipitated in the ferrite matrix, and the strength and toughness matched well; when the tempering temperature rose above 670 °C, hard and brittle plate-martensites formed at the grain boundary, leading to the tensile strength of the experimental steel increased, while the low-temperature impact toughness significantly decreased and the yield strength also declined due to the disappearance of the finely dispersed second phase particles in the matrix.

**Keywords:** SA738 Gr.B; high-temperature tempering; mechanical properties; microstructure

## 1. Introduction

In recent years, nuclear energy, as a clean energy source, has received considerable attention from countries in the context of increasingly tight world energy supplies and worsening environmental problems. Nuclear safety is an integral part of national security [1]. The typical nuclear power reactor is AP1000, designed by Westinghouse [2]. The structure of the AP1000 nuclear reactor is a double-layer envelope structure composed of steel and reinforced concrete, which is different from the traditional prestressed concrete envelope structure [3]. Therefore, the unconventional design of the AP1000 containment has put forward higher requirements of strength and low-temperature toughness. To meet the higher standard [4], the primary material of its containment is an SA738 Gr.B steel plate in the American Society of Mechanical Engineers (ASME) SA738/SA738 M standard [5], according to the material specification, SA738 Gr.B steel plates shall have a minimum impact energy of 27 J either at −45 °C in the quarter of the thickness direction. To obtain a better combination of strength and low-temperature impact toughness, the typical heat treatment process of SA738 Gr.B is quenching and tempering [5]. For the quenching and tempering process of SA738 Gr.B steel, relevant scholars have carried out related research: Han et al. [6] and Sun [7] studied the ferrite matrix after quenching and tempering and found that the increase in tempering temperature coarsened the grains, and with the increase in the soft and ductile phase ferrite content, the strength of SA738 Gr.B steel decreased and the impact energy increased. Bi et al. [8] observed the precipitated phases after tempering at 630 °C by TEM; it was found that a large number of carbides with various shapes were dispersed and precipitated between the laths, hindering the growth of grains and inhibited the movement of dislocations. Zhang [9] et al. used the master curve (MC) method to modify the ductile–brittle transition zone and fracture toughness of the tempered SA738 Gr.B steel, which proves that it has sufficient toughness reserve at a low temperature. However, a systematic and comprehensive study on

the brittle temperature range and the mechanisms of strengthening and toughening of SA738 Gr.B steel is still lacking.

In this paper, the effects of tempering temperature on the strength and toughness of SA738 Gr.B nuclear power steel are systematically studied by comparing the strength and low-temperature impact energy at different tempering temperatures and clarifying the evolution of the microstructure combined with structural characterization methods such as SEM and TEM, and thermodynamic software calculations.

## 2. Material and Methods

The material used in the research was rolled SA738Gr.B steel, and the thickness was 46mm. Its chemical composition is shown in Table 1. Table 1 also lists the chemical composition range of SA738 Gr.B steel in the ASME standard [5]. The steel plate was cut in the quarter of the thickness direction, and several samples with dimensions of $12 \times 40 \times 120$ mm$^3$ were taken for heat treatment. The samples were austenitized at 900 °C for 30 min, followed by water quenching to room temperature, and then tempered at 630, 650, 670, 690, and 710 °C for 60 min, respectively. Figure 1 is the SEM micrograph of SA738 Gr.B steel after quenching, through which we can conclude that the microstructure of as-quenched steel is lath martensite.

**Table 1.** Chemical compositions of the steel SA738 Gr.B in ASME standard and this experiment (mass fraction, %).

| Element | C | Mn | P | S | Si | Ni | Cr | Mo | Nb | V | Ti | Fe |
|---|---|---|---|---|---|---|---|---|---|---|---|---|
| Standard | ≤0.2 | 0.9–1.5 | 0.008 | 0.005 | 0.15–0.55 | ≤0.6 | ≤0.3 | ≤0.3 | | Total ≤ 0.08 | | Bal. |
| Experimental | 0.14 | 1.55 | 0.008 | 0.001 | 0.25 | 0.55 | 0.23 | 0.28 | | Total = 0.077 | | Bal. |

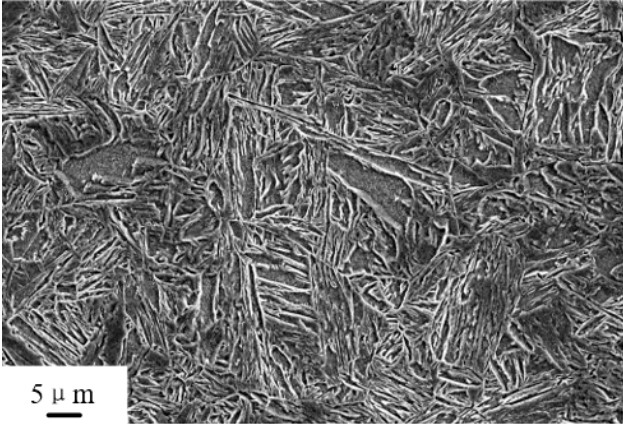

**Figure 1.** SEM image of as-quenched SA738 Gr.B steel.

After heat treatment, Charpy V-notch specimens with dimensions of 10 mm × 10 mm × 55 mm$^3$ were prepared. The V-notch was parallel to the rolling direction. Charpy impact tests were performed at −45 °C on an Instron Dynatup 9250 instrumented impact machine (Norwood, MA, USA) according to the working condition of the steel. Three specimens were tested for each heat treatment process in order to guarantee reproducibility. Specimens for the strength tests with a 5 mm diameter and 50 mm gauge length were prepared according to GB/T 228–2010 [10].

Samples for metallographic analysis were prepared by wire-cut electrical discharge machining. The microstructures were observed by ZEISS ULTRA 55 Field Emission Scanning Electron Microscopy (FESEM, Carl Zeiss AG, Jena, Germany) after grinding, polishing, and etching with 4% nital solution. A Tecnai G2 F20 transmission electron microscope (TEM, FEI, Hillsboro, OR, USA) was also used to observe the dislocations and phases. The distribution of alloy elements was analyzed by using JEOL JXA-8530F field emission electron probe (EPMA, Jeol Co. Ltd., Kyoto, Japan). The crystal structures of the particles were identified by Digital Micrograph software (Gatan, Pleasanton, CA, USA). The diagram

between mass fraction and temperature of all the phases and the trend of the precipitation type and quantity of the carbides were calculated by JMatPro software (Version 6.0, General Steel module, Sente Software Ltd., Guildford, UK) according to the chemical composition of SA738 Gr.B.

## 3. Results

### 3.1. Mechanical Properties

Figure 2a shows the strength and elongation change of SA738 Gr.B steel after tempering at different temperatures for 60 min. It can be known that the tensile strength of SA738 Gr.B steel decreases from 659 MPa at 630 °C to 611 MPa at 670 °C; when tempered at 690 and 710 °C, the tensile strength increased, reaching a maximum of 698 MPa. The yield strength gradually decreased from 581 MPa at 630 °C to 498 MPa at 710 °C. The elongation reached the maximum of 20.8% at 650 °C, and gradually decreased with the increase in tempering temperature, and the final is 17.66% at 710 °C. Figure 2b shows the impact toughness of SA738 Gr.B steel at −45 °C. The impact energy of the steel maintained at 250 J at 630 to 650 °C; the impact energy dropped to 203 J at 670 °C; when the tempering temperature is 690 °C, the impact energy decreased sharply to 83 J, and 42 J at 710 °C. In summary, when the tempering temperature is lower than 670 °C, SA738 Gr.B steel has better comprehensive mechanical properties.

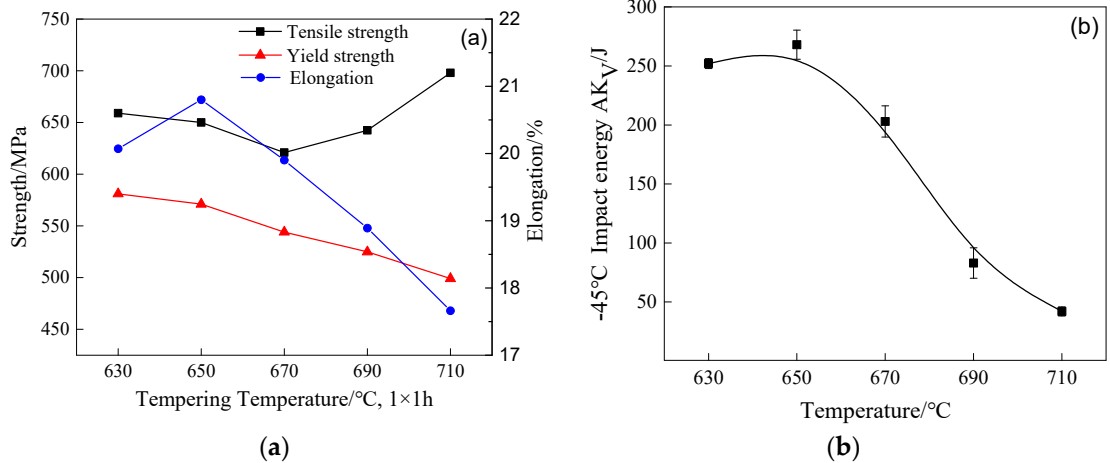

(**a**)                    (**b**)

**Figure 2.** Strength, elongation (**a**), and impact energy (**b**) of tempered steel at different temperatures.

### 3.2. Microstructure of the Tempered Steel

Figure 3 shows the SEM morphology images of SA738 Gr.B steel at different tempering temperatures. After tempering at 630 °C, the microstructure of the steel was tempered sorbite, the ferrite matrix had a polygonal or a lath structure (Figure 3a), and the width of the lath structure was less than 1 μm. A large number of second-phase particles precipitated on the matrix or laths diffusely (Figure 3b), with a size of less than 200 nm. When tempered at 650 °C, the microstructure was roughly the same as 630 °C. The size of the second-phase particles increased, with a maximum size of more than 200 nm (Figure 3c,d). When tempered at 670 °C, a small amount of island-like structures could be observed at the grain boundaries, with the size between 0.2 and 1 μm (Figure 3e), and the number of second-phase particles on the ferrite matrix was reduced (Figure 3f). The island-like structure continued to precipitate along the grain boundary when tempered above 690 °C; the volume fraction gradually increased, and interconnected to become the main phase of the microstructure (Figure 3g,h). The average proportion of the island-like structure in the tempered structure at 670–710 °C was 5.1%, 18.6%, and 34.9%, respectively.

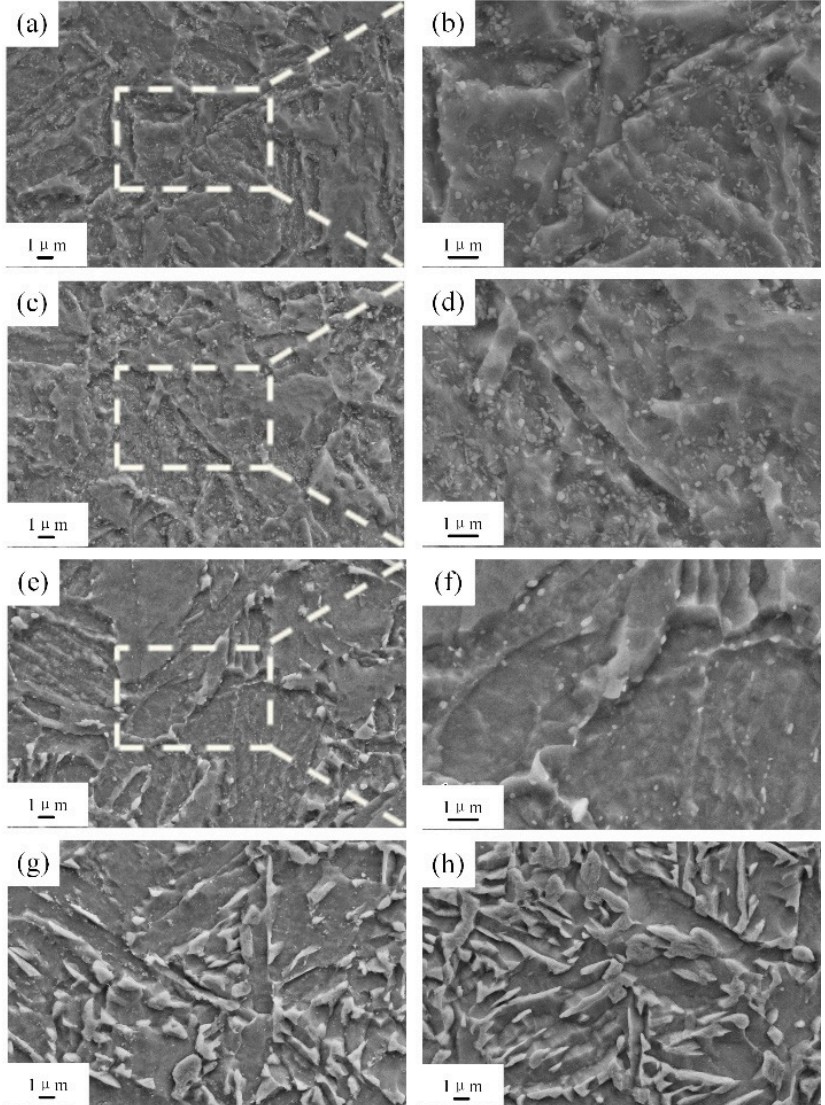

**Figure 3.** SEM images of SA738 Gr.B steel tempered at different temperatures: (**a**,**b**) 630 °C; (**c**,**d**) 650 °C; (**e**,**f**) 670 °C; (**g**) 690 °C; (**h**) 710 °C.

The fracture morphology of the SA738 Gr.B steel impact specimen is shown in Figure 4. It could be seen that when the tempering temperature was 630–670 °C, the corresponding fracture was a typical dimple-type fracture morphology, as shown in Figure 4a–c. The sections were all composed of plastic pits. A large plastic deformation occurred before the fracture, and the corresponding impact energy was relatively high; when the tempering temperature was 690 and 710 °C, as shown in Figure 4d,e, the fracture morphology was a cleavage plane, whose typical river pattern could be observed, and there were block craters (red dashed lines) and secondary cracks; that is, the original block structure was the source of cracking, corresponding to the island structure in the SEM (Figure 3). The size of the plane was smaller at 690 °C, and the river pattern was zigzagged and intertwined, which indicated that the unit expansion path was small. After the microcracks blocked along the cleavage plane and the expansion direction changed, it required more energy. The crack initiation area was large at 710 °C, so the corresponding impact energy was the lowest.

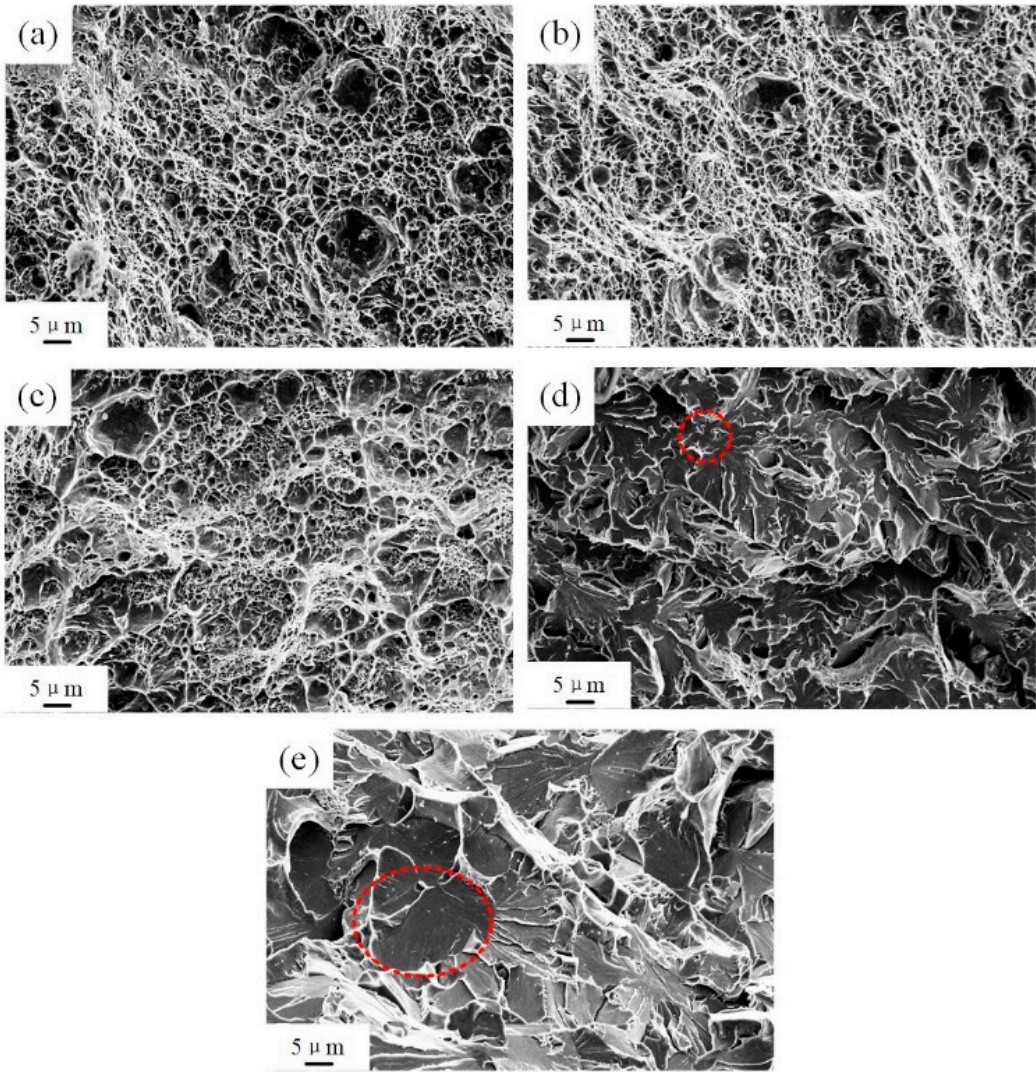

**Figure 4.** Fracture surface morphologies of SA738 Gr.B steel tempered at 630 °C (**a**), 650 °C (**b**), 670 °C (**c**), 690 °C (**d**), 710 °C (**e**) after a Charpy impact test at −45 °C.

Figure 5 is TEM micrographs of SA738 Gr.B steel after tempering at 630 and 710 °C. When the steel tempered at 630 °C, part of the laths could be observed. The second phase particles precipitated along with favourable positions such as grain boundaries, lath boundaries, or the matrix as shown in Figure 5a; at 650 °C, the morphology of the lath disappeared, and the carbide distribution was the same as 630 °C. The size of the large-sized precipitates ranged from 100 to 300 nm, as shown in Figure 5a,b. Differently, it was strip-like or spherical; the small-size carbides were less than 100 nm, as shown in Figure 5c, and it was square and distributed in the matrix. By using JmatPro thermodynamic software (Figure 6) and Energy Dispersive Spectrometer (Figure 7), the large-sized precipitates were $M_3C$-type alloy carbides (M = Fe, Mn, Cr), and the small-sized precipitates were MC-type alloy carbides (M = Ti, Nb, Cr, V). Because of the low content and small size of MC carbides, the precipitation strengthening and the damage to toughness are limited, so we could ignore the discussion of MC carbides in this paper.

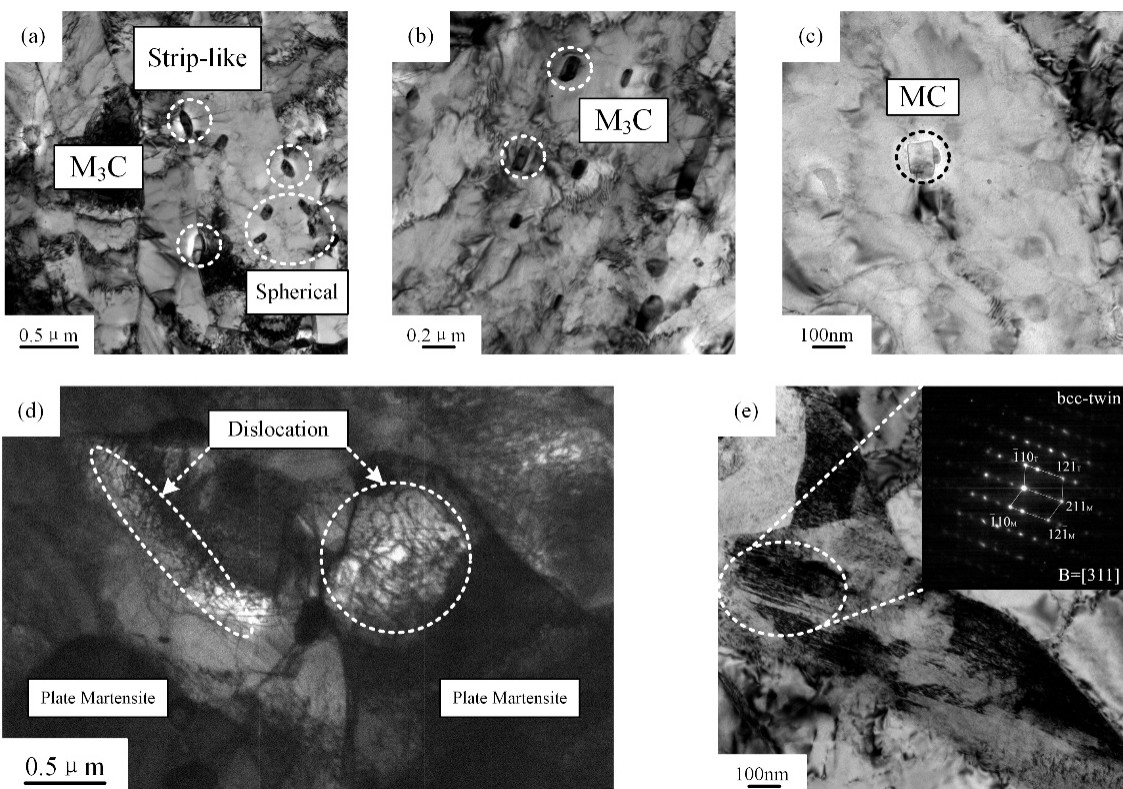

**Figure 5.** TEM images and selected diffraction spots of SA738 Gr.B steel after tempering at different temperatures: (**a**) 630 °C; (**b**,**c**) 650 °C; (**d**) 690 °C; (**e**) 710 °C.

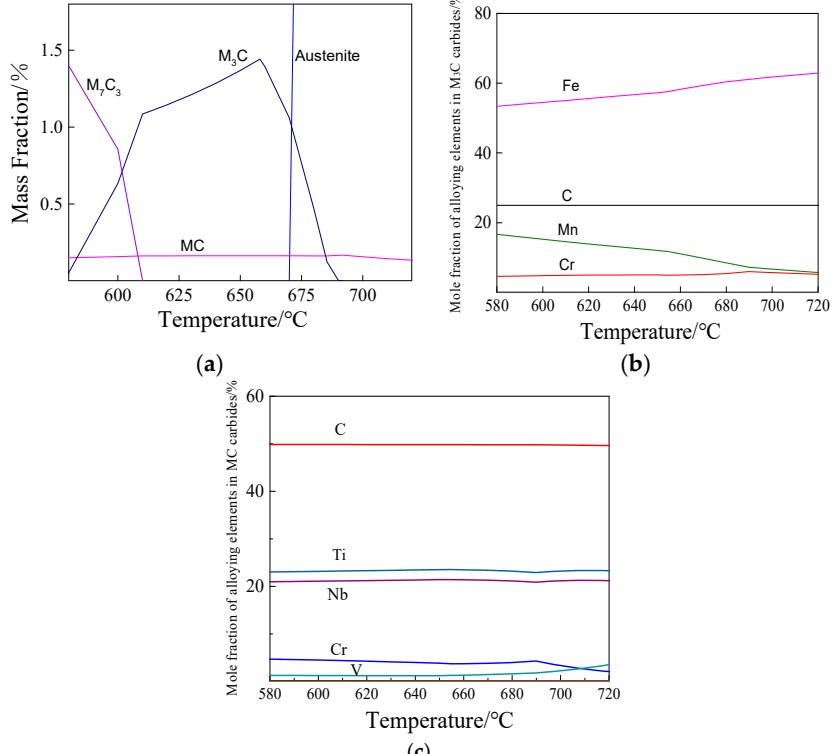

**Figure 6.** Thermodynamic calculation of the second phase particles. (**a**) The second phase precipitates in the tempering temperature range; (**b**) the composition of the M3C; (**c**) the composition of the MC.

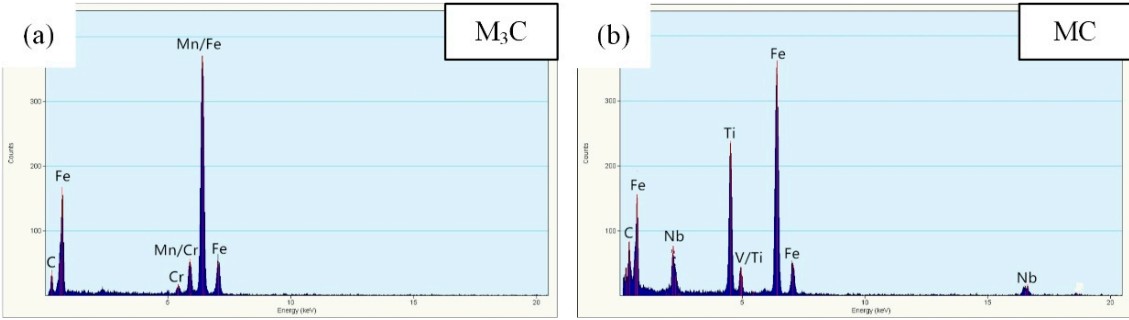

**Figure 7.** Energy Dispersive Spectrometer(EDS) analysis of precipitates in the tempered SA738 Gr.B steel. (**a**) $M_3C$; (**b**) MC.

At 690 and 710 °C, as shown in Figure 5d, almost no precipitates could be observed, but a large area of black island-like structure could be observed, and its size was more than 500 nm. Corresponding to the island-like structure, the surrounding stress field was strong due to the phase change, so high-density dislocation networks distributed on the surrounding ferrite matrix; Figure 5e is the morphology of the island-like structure and the corresponding selected area electron diffraction spectrum. The diffraction pattern determined that it was plate martensite with a bct-twin structure. The thickness of the twin layer was about a few nanometers and was relatively uniform.

## 4. Discussion

### 4.1. Three Different Tempering Temperature Ranges

According to the microstructure of SA738 Gr.B steel after high-temperature tempering, the tempering temperature can be divided into three different ranges:

1. Tempering at 630–650 °C: There was little change in the strength and toughness. The changes in the microstructure mainly included the precipitation and growth of $M_3C$ carbides and the polygonalization of ferrite.

2. Tempering at 670 °C: The strength and toughness were reduced. A small amount of island-like plate-martensite began to form on the grain boundary, and the size of $M_3C$ carbides in the matrix increased. In this process, the larger $M_3C$ particles grow at the expense of the smaller particles. This further reduces the surface energy and is accomplished by the diffusion of carbon atoms through the iron matrix [11].

This range continued the changes of the ferrite matrix and carbides from the previous stage. Although a small amount of hard and brittle phases generated, but the amount was not enough to increase the strength; so the overall strength of the steel was reduced. Correspondingly, the softened matrix interacted with the hard and brittle plate-martensite, and the impact toughness did not decrease much. The transition point of tempering temperature of 670 °C can be used.

3. Tempering at 690–710 °C: During this range, the ferrite was completely polygonal, and the lath morphology completely disappeared. A large amount of plate-martensite precipitated and grew along the grain boundaries during cooling after tempering. $M_3C$ carbides may precipitate in the initial stage, but as the tempering temperature increased, carbon and alloys elements would gradually be enriched in austenite with a higher solid solubility. This can also be fully proved in the calculation of thermodynamic software, as shown in Figure 6a. Finally, ferrite, plate-martensite, and retained austenite structures can be obtained.

*4.2. Mechanism of Effect of Plate Martensite on Properties*

During the transformation of austenite to plate-martensite, the volume expansion due to the lattice expansion; the expanded plate-martensite compressed the ferrite matrix; so the stress-induced high density dislocation effectively hinders slippage [12], resulting in significantly improved tensile strength.

Additionally, according to the Ashby–Orowan mechanism [13], the yield strength $\sigma_{ppt}$ due to precipitation strengthening can be expressed as follows:

$$\sigma_{ppt} = \frac{0.8MGb}{2\pi L \sqrt{1-v}} \ln\left(\frac{x}{2b}\right) \tag{1}$$

where

$$L = \sqrt{\frac{2}{3}\left(\sqrt{\frac{\pi}{f}}-2\right)}r \tag{2}$$

$$x = 2\sqrt{\frac{2}{3}}r \tag{3}$$

so the $\sigma_{ppt}$ can be expressed as another form:

$$\sigma_{ppt} = \frac{a + c\ln r}{\left(\sqrt{\frac{\pi}{f}}-2\right)r} \tag{4}$$

Here, M is the Taylor constant, G is the shear modulus, v is the Poisson's ratio, b is the Berkeley vector mode, r is the size of the precipitated phase, f is the volume fraction of the precipitated phase, a and c are constants. From Formula (4), it can be seen that the precipitation strengthening effect is mainly related to f and r; that is, the larger the volume fraction of the precipitated phase, the smaller the r, the higher the strength. The amount of $M_3C$ carbides in the matrix decreased sharply; almost no carbide precipitated at 710 °C, and the precipitation strengthening effect in the ferrite matrix was greatly reduced, resulting in a continuous decrease in the yield strength of the steel.

When there are twin substructures in martensite, the number of slip systems reduces, and the dislocations need to be in a "Z" shape when passing through the twins, which increases the resistance to deformation and leads to stress concentration. So large-sized martensite twins can promote crack growth and severely deteriorate the low-temperature toughness of steel [14]. At the same time, EPMA was used to analyze the plate-martensite. As shown in Figure 8, it was found that the phosphorus element segregated on the prior austenite grain boundary. Takayama et al. [15] proposed a method for calculating the ductile–brittle transition temperature of steel using experiments and Taylor series expansion. The results showed that the ductile–brittle transition temperature of steel increased with the increase in the degree of phosphorus segregation, so the phosphorus segregation seriously affects the low-temperature toughness of steel.

When the matrix was subjected to impact load, plate-martensite hindered the movement of dislocations in the matrix, causing dislocation plugging, as shown in Figure 5. The plugged dislocation, in turn, exerted stress on the plate-martensite [16]. The stress interaction caused the plate-martensite to break or caused the surrounding ferrite to form microcracks due to stress concentration. Therefore, microcracks that met the Griffith condition nucleated on the plate-martensite or the interface of the martensite-matrix [17], which caused cleavage fracture. Observation of secondary cracks on the fracture side section of the impact specimen by SEM also verified this view, as shown in Figure 9.

In summary, plate-martensite is the preferred location for crack initiation and propagation [12] and is the direct cause of the low-temperature impact toughness reduction. The schematic diagrams of the microstructure evolution, precipitation behaviour, and crack propagation path of SA738 Gr.B steel at different tempering temperatures are shown in Figure 10.

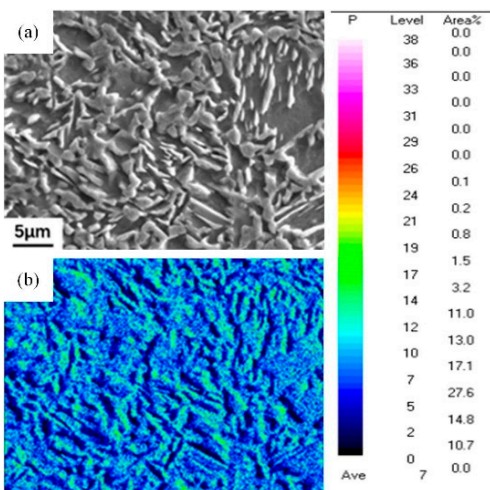

**Figure 8.** EPMA analysis of the segregation of phosphorus. (**a**) Microstruture; (**b**) segregation of phosphorus.

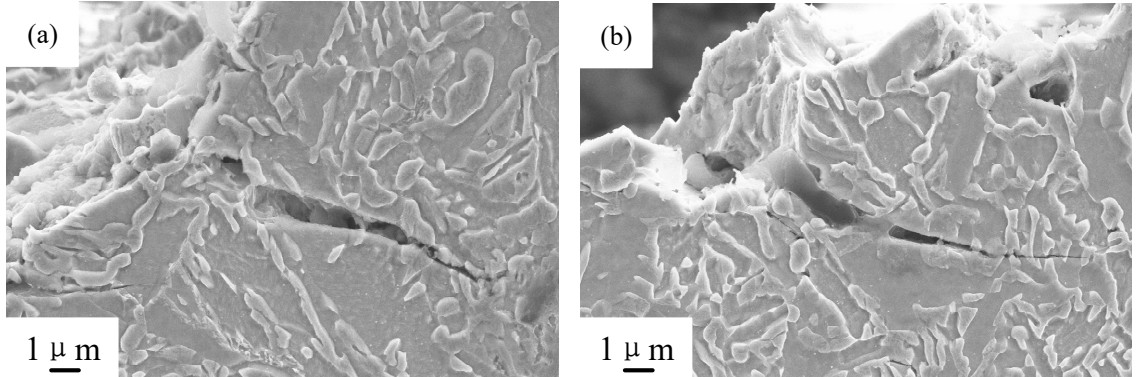

**Figure 9.** SEM images of the cross-sectional area of the Charpy impact specimens fractured at −45 °C for the SA738 Gr.B steel tempered at 690 °C (**a**), 710 °C (**b**).

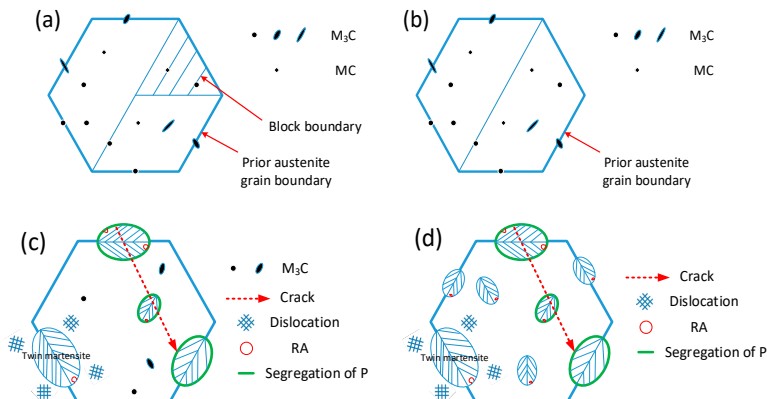

**Figure 10.** Schematic representation for microstructure and precipitates of SA738 Gr.B under different tempering temperatures: (**a**) 630 °C; (**b**) 650 °C; (**c**) 690 °C; (**d**) 710 °C.

### 4.3. Formation Mechanism of Plate Martensite

Regarding the formation mechanism of twin-martensite in steel, it is generally believed that the martensite twins are caused by the shear transformation of the austenite [18]. As carbon content in

austenite increases, the substructure of martensite transitions from dislocation type to twin type [19]. Additionally, Huang et al. [20] believed that the substructure of martensite also evolved with the decrease in the phase transition temperature. Dislocation-type martensite generated at higher temperatures and twin-type at lower temperatures.

The SA738 Gr.B steel used in this experiment was a type of low-carbon steel: plate martensite will not form. In fact, through the calculation of JMatPro software, the $A_{C1}$ temperature of the experimental steel was 675 °C. Heat treatment above 675 °C did not temper in the traditional sense; so a certain amount of austenite was formed, and during the cooling, only the austenite had phase transformation. JMatPro thermodynamic software was also used to calculate the main chemical component content and $M_s$ temperature of the austenite at various temperatures, as shown in Table 2. It can be seen that the austenite phase formed in the tempering stage could be considered to be high-carbon steel, ensuring the composition fluctuation to form the plate-martensite. Besides, Kelly et al. [21] discovered that in iron-carbon alloys, phase transition twins easily formed when the $M_s$ point was lower than 300 °C due to the increase in the mass fraction of carbon. Mn, Ni, Mo, and Cr can not only improve the hardenability of steel but also increase the tendency to form plate-martensite [22]. As can be seen from Table 2, the austenite under this composition had a lower $M_s$ point and a higher alloy content. Under the continuous cooling condition, due to the small size of the sample, the cooling rate was fast in the air, and the distortion energy was constantly increasing. If martensite grows in the direction of symmetry of the crystal plane during the growth process, the strain energy can be reduced and adjusted to meet the energy fluctuation. This is also an essential reason for the formation of martensite twins [15].

**Table 2.** Elemental mass fraction (%) for equilibrium austenite phase of SA738 Gr.B steel at different tempering temperatures and $M_s$ (°C) point.

| Temperature/°C | C | Mn | Ni | Cr | Mo | $M_s$ |
|---|---|---|---|---|---|---|
| 690 | 0.52 | 3.80 | 1.15 | 0.36 | 0.19 | 161.6 |
| 710 | 0.42 | 3.23 | 1.03 | 0.34 | 0.21 | 175.7 |

Ahn et al. [23] found that adding intercritical heat treatment which performed in the ferrite+austenite ($\alpha + \gamma$) temperature region between the quenching and tempering processes of SA508-3, which is also a type of nuclear pressure vessel steel, can effectively improve the toughness, and its matrix composition (Fe-0.21C-1.24Mn-0.25Si-0.88Ni-0.21Cr-0.47Mo) (wt%) is similar to the SA738 Gr.B steel. However, according to the analysis of the current results, tempering at 690 and 710 °C is equivalent to a intercritical heat treatment. Carbon atom distribution of ferrite-martensite was uneven compared with that of lath martensite after quenching. After tempering again, carbides were unevenly precipitated near the island-like structure, which inevitably affected the properties of the steel. Wang et al. [24] also proved that their performance was poor after tempering SA738 Gr.B steel with ferrite-martensite dual-phase structure. Therefore, in industrial production, the tempering temperature should be strictly controlled to achieve a well-fit strength and toughness.

## 5. Conclusions

In this paper, in order to achieve suitable strength and toughness for SA738 Gr.B nuclear power steel, five tempering experiments with different processes were conducted, and the mechanical properties and microstructure were tested and characterized. The following conclusions were drawn from the test results:

1. The best tempering temperature range of SA738 Gr.B steel is 630–670 °C, and it has a better combination of strength and low-temperature impact toughness.

2. When the tempering temperature of SA738 Gr.B steel is lower than 670 °C, the microstructure of SA738 Gr.B steel still maintains the lath morphology. The precipitated particles of the second phase are fine $M_3C$ and MC which are dispersed; so the steel has a better combination of strength and

toughness. When the tempering temperature was higher than 670 °C, a large amount of carbon-rich austenite formed during the tempering process, and most of them transformed into hard and brittle plate martensite which is favourable for crack growth, leading to the sharp decrease in impact energy at −45 °C; the yield strength decreased due to the disappearance of the fine dispersed second particles in the matrix.

**Author Contributions:** Conceptualization, Y.L. and S.Z.; investigation, S.Z.; data curation, C.Z. and M.S.; writing—original draft preparation, S.Z.; writing—review and editing, Y.L., C.Z., M.S., and Z.J.; funding acquisition, Y.L. All authors have read and agreed to the published version of the manuscript.

**Funding:** This research was supported by the China National Key R&D Program during the 13th Five-year Plan Period (Grant No. 2017YFB0305301).

**Conflicts of Interest:** The authors declare no conflict of interest.

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
