# Peer review of "Effects of High-Temperature Tempering on Mechanical Properties and Microstructure of SA738 Gr.B Steel"

_metals, doi:10.3390/met10091207_

Round 1
Reviewer 1 Report
1/ row 52. The authors state that the product size was 46 mm. What dimension it is? Later they write that several samples were, cute from it for heat treatment. After that the samples for Charpy V-notch were prepared in size of 10x10x55 mm. The value of 46 mm don’t describe the geometry of the material fully. What was the length, width and thickness of the material?
2/ What was the cooling rate after tempering?
3/ What was the procedure for metallography sample preparation? Were the samples grinded, polished and etched? If yes what kind of etchant was used? Or you electro-polished the samples?
4/ In case of JMatPro what database and which module was used for the calculations.
5/ Row 79 the sentence “the temperature increase below 670°C”. Have can the temperature increase below something? Looking at the rest of the sentence it should be “higher” or “upper” then 670°C, because the authors compare that temperature with 710°C.
The same for the comparison. The “tensile strength decrease with increased temperature below 670”. But in text authors compare 690 with 710°C. What is the tensile strength at 670°C? If the strength decrease when the tempering is in the higher temperature then they should present the TS value for this temperature to have order in their description in the text.
6/ Similar situation in case of impact toughness the authors compare the 630, 650, 690 and 710, and state that the drop in the impact toughness is presented at temperatures higher then 670°C. What is the impact toughness at 670°C? It should be explained it the text. According to figure 1 the drop of impact toughness at 670°C is around 50J. But in text the authors for some reason miss this temperature in the description in text.
7/ Row 112 please put the exact figures in the description in text. There is SEM (fig.2) it should be SEM (fig.2e-h), because these are the only microstructures with island like structures.
8/ In the section of SEM analysis the authors state that the microstructure of 630 and 650 is the same. Only the secondary phases size increased. But in TEM analysis they stated that in case of 650 (row 123) that morphology of the laths disappeared. So is the structure the same or different?
9/ In discussion in point 4.1 The authors describe only M3C particles. However earlier they prove the presence of MC particles. Why this is not discussed in this point?
10/ In conclusions the authors once again stat the at 630 and 650 have the lath like structure and MC particles are presented too.
Author Response
Dear Reviewers:
Thank you for your letter and for the reviewers’ comments concerning our manuscript entitled “Effects of high-temperature tempering on mechanical properties and microstructure of SA738 Gr.B steel” (ID:metals-905981 ). Those comments are all valuable and very helpful for revising and improving our paper, as well as the important guiding significance to our researches. We have studied comments carefully and have made correction which we hope meet with approval. Revised portion are marked in red in the paper. The main corrections in the paper and the responds to the reviewer’s comments are as flowing:
1. Response to comment: The size of the product
Response: We are sorry for this mistake.The thickness of the product is 46mm, both length and width are sufficient for sampling. The steel plate was cut in the quarter of the thickness direction, and several samples with dimensions of 12mm × 40 mm × 120 mm were taken for heat treatment.
2. Response to comment: The cooling rate after tempering
Response: The tempering cooling rate measured by the thermocouple is 0.72℃/s on average .
3. Response to comment:The procedure for metallography sample preparation.
Response:The sample was prepared after grinding, polishing, and etching with 4% nital solution. We are sorry I spelled the word nital as natal and I have corrected.
4. Response to comment:JMatPro calculation
Response:We are very sorry for our negligence of the version. JMatPro software is version 6.0, the module is Step Temperature,I have added to the paper.
5&6 Response to comment:Tensile strength and toughness for 670℃
Response:We have re-written this part according to the Reviewer’s suggestion and we have added the data for 670℃
7.Response to comment:The exact figures in the SEM.
Response:The average proportion of the island-like structure in the tempered structure at 670℃ ~ 710℃ was 5.1%, 18.6%, and 34.9%, respectively.The data was calculated by image pro plus software and added in the line 102 to 103.
8.Response to comment:The microstrutre of 630℃ and 650℃.
Response:In SEM, due to insufficient magnification, it is difficult to distinguish the laths and polygonal structures. It is indeed not rigorous enough. We have modified,the microstructures of 630℃ and 650℃ is roughly the same.
9&10:Response to comment:MC carbides and M3C
Response:We are very sorry for our negligence of the MC carbides.The size of MC carbides is small, between 50~100nm, and the content of microalloying elements of the this steel is low, the number of MC carbides is also very small, so the carbide that really plays a role in performance is coarsened M3C , So we only mentioned the observation here, but did not focus on the discussion of MC.
Special thanks to you for your good comments.
Reviewer 2 Report
The paper addresses a new and appealing research topic, the approach developed is quite clear. In order to enhance the quality of this paper, the following issues have to be addressed/improved before considering the paper for the publication:
General: In many places (for example 79-84, 90-101) there is a space between a temperature value and degrees e.g. 710 °C. Please remove these spaces. The similar problem is with the MPa (have a look: 79, 81). Please unify.
22. “(…) as a clean energy source (…)”: It is true that nuclear energy has many advantages, but it is not “a clean energy source”.
68. “(…) erosion (…) natal (…)”: By “erosion” you mean “etching”? Also, it should not be “natal” instead of “natal”?
74. “composition of SA738 Gr.B”: Add a dot at the end of the sentence.
88. It should be “3.2” instead of “3.1”.
99-101. “The average proportion of island-like structure (…)”: How these percent values were determined? It is not so clear to evaluate it by the reader because Fig. 2g and Fig. 2h have different scales (what also causes a problem with the interpretation of Fig. 3d and Fig. 3e). Please explain the methodology of this measurement/calculation and also check the scales in Fig. 2 (In my opinion, you have put a wrong scale on Fig. 2h).
138. “(…) SA738 Gr.b (…)”: It should be a capital “B”.
144. The presented results of EDS are of poor quality.
150. “(…) there is little changein (…) thoughness .The (…)”: It should be “change in” and “toughness. The (…).
171. “(…) squeezed (…)”: I think that instead of “squeeze” you mean “compress”.
239. “(…) due to the small size of the sample (…)”: This is the main problem. The presented results are clear and the formation of martensite twins and their effect on the properties of SA738 Gr.B is well-described, but it corresponds only to the small samples (thickness of 12mm). The influence of investigated heat treatments for bigger elements (e.g. 50mm) can differ significantly – for example in terms of martensite-twins participation. Additionally, the martensite participation can change in the cross-section of the sample for a lower cooling rate in the element interior (with the formation of simple ferrite-perlite microstructure?). For this reason, some sentences in your manuscript such as 12-13 “The results showed that the suitable tempering temperature range is 630℃ ~ 670℃” can be misleading. I would suggest adding the information about sample thickness in such sentences.
259. “(…) low-temperature impact toughness”: Add a dot at the end of the sentence.
Author Response
Dear Reviewers:
Thank you for your letter and for the reviewers’ comments concerning our manuscript entitled “Effects of high-temperature tempering on mechanical properties and microstructure of SA738 Gr.B steel” (ID:metals-905981 ). Those comments are all valuable and very helpful for revising and improving our paper, as well as the important guiding significance to our researches. We have studied comments carefully and have made correction which we hope meet with approval. Revised portion are marked in red in the paper. The main corrections in the paper and the responds to the reviewer’s comments are as flowing:
1.Response to comment:General
Response:We have made correction according to the Reviewer’s comments.
2.Response to comment:line 22
Response:Nuclear energy is complicated. From the perspective of conventional energy, nuclear energy does not produce sulfur, nitrogen compounds and special carbon dioxide. From this perspective, nuclear energy is a clean energy.
But the biggest problem with nuclear energy is nuclear waste, reprocessing and nuclear accidents. If these problems are not solved properly, the use of nuclear energy will be greatly restricted.
3.Response to comment:line 68
Response:We are very sorry for our incorrect writing,We have corrected in the paper.
3.Response to comment:line 74
Response:We have made correction according to the Reviewer’s comments.
4.Response to comment:line 88
Response:We have made correction according to the Reviewer’s comments.
5.Response to comment:line 99-101
Response:We are very sorry for our mistake of the scales in Fig.2h and have corrected. The methodology of this measurement is painting the island-like structure in a high-contrast color (for example, red), and then use the Image
Pro Plus software counts the proportion of the red part.
6.Response to comment:line 138
Response:We have made correction according to the Reviewer’s comments.
7.Response to comment:line 144
Response:It is really true as Reviewer suggested that we have replaced the EDS with a clearer picture.
8.Response to comment:line 150
Response:We have made correction according to the Reviewer’s comments.
9.Response to comment:line 171
Response:We have made correction according to the Reviewer’s comments.
10.Response to comment:line 239
Response:It is indeed misleading. We have carefully considered your suggestion and finally deleted the sentence The results showed that the suitable tempering temperature range is 630℃ ~ 670℃ in the abstract.
11.Response to comment:line 259
Response:We have made correction according to the Reviewer’s comments.
Special thanks to you for your good comments.
Reviewer 3 Report
The authors investigated microstructure and mechanical properties of SA738 Gr. B nuclear power steel after tempering at 630 °C~ 710 °C. Tempering within the range of 630 °C~ 670 °C was identified as suitable tempering range. The authors related it to the huge amount of M3C and MC carbides, that were dispersed and precipitated in the ferrite matrix and increased strength and toughness. When the tempering temperature rose above 670°C they found hard and brittle twin-martensite, formed at the grain boundaries, leading to increasing tensile strength of the experimental steel but significant decrease in low-temperature impact toughness.
This work is a systematic study on the effect of tempering temperature on strength and toughness of SA738 Gr. B 46 nuclear power steel. The different phases within the microstructure were identified by means of scanning as well as transmission electron microscopy and could be related to strength and low-temperature impact energy at different tempering temperatures The subject matter is suitable for publication in Metals but with respect to content and figures there are some open questions:
Line 36: How can grain boundaries coarsen? I think you mean grains.
Line 40: How can grain boundaries grow? I think you mean grains again.
Lines 93-94: Contradiction: “When tempered at 650 ℃, the microstructure is the same as 630 ℃” and in the following line: “The size of the second-phase particles increased, with a maximum size of more than 200 nm (Figure 2c-d).”
Figure 2: The particles are hardly visible
Figure 8b: Legend is not readable, it is much too small
Line 190: How can slip systems reduce? Do you mean that the number of slip systems is reduced?
Conclusion: Introductory sentence is missing. With respect to the cover story no information about the meaning of the results for this special application is given. How can you transfer the results of very small laboratory specimen to real, large constructions? Could you please add a short comment?
References to the related work are exclusively from Asian authors (except three quite old publications from 1971, 2001 and 2006 respectively). Is there really no current literature from international authors?
The manuscript exhibits some linguistic and grammatical mistakes and needs revision.
Author Response
Dear Reviewers:
Thank you for your letter and for the reviewers’ comments concerning our manuscript entitled “Effects of high-temperature tempering on mechanical properties and microstructure of SA738 Gr.B steel” (ID:metals-905981 ). Those comments are all valuable and very helpful for revising and improving our paper, as well as the important guiding significance to our researches. We have studied comments carefully and have made correction which we hope meet with approval. Revised portion are marked in red in the paper. The main corrections in the paper and the responds to the reviewer’s comments are as flowing:
1.Response to comment:line 36
Response:We have made correction according to the Reviewer’s comments.
2.Response to comment:line 44
Response:We have made correction according to the Reviewer’s comments.
3.Response to comment:line 93-94
Response:We are very sorry for our incorrect writing .We have changed the sentence to the microstructure is roughly the same as 630℃.
4.Response to comment:Fig.2
Response:It is really true as Reviewer suggested that we have replaced the SEM with a clearer picture.
5.Response to comment:Fig.8b
Response:We think you mean Fig.7 EPMA? It is really true as Reviewer suggested that we have replaced the EPMA with a clearer picture.
6.Response to comment:line 190
Response:We are very sorry for our incorrect writing,we have changed it to the number of slip systems
7.Response to comment:Conclusion
Response:We are very sorry for our negligence of the introductory sentence and we have added.
The specimen size is indeed relatively small, which is quite different from the industrial production. According to the ASME standard, the -45℃ impact energy at 1/4 of the thickness shall not be less than 27J. A steel production company once provided us with a quenched 100mm industrial steel plate. We also conducted tempering experiments in the laboratory (the tempering time increased according to the thickness). The results showed that as long as the temperature exceeds 690℃, A certain amount of twinned martensite will be produced at 1/4, its quantity is not large, but it will also affect the mechanical properties of the steel. The structure of which is bainite + twinned martensite. Therefore, the small specimen also has certain guiding significance for the industrial production.
We put the photos of the microstructure in the attachment.
As Reviewer suggested that references to the related work are exclusively from Asian authors.
Indeed, the proportion of Asian authors in our reference is relatively high, because SA738 Gr.B nuclear power steel is a confidential project. Some foreign researchers keep their results confidential after successful development and have not published them in journals. Therefore, during the process , we did not find much information about SA738 Gr.B nuclear power steel. I really appreciate your understanding
8.Response to comment:The manuscript exhibits some linguistic and grammatical mistakes and needs revision.
Response:We have made correction according to the Reviewer’s comments.
Special thanks to you for your good comments.

Round 2
Reviewer 1 Report
Just two samll things are left to correct. Below I put you answers and what i find that could be improven.
Response to comment:JMatPro calculation
Response:We are very sorry for our negligence of the version. JMatPro software is version 6.0, the module is Step Temperature,I have added to the paper.
Step temperature is kind of calculations that you can make in JMatPro, what I mean by module is, if you used general steel module or stainless steel module…? Depending on the module, the software use different model for the calculations.
9&10:Response to comment:MC carbides and M3C
Response:We are very sorry for our negligence of the MC carbides.The size of MC carbides is small, between 50~100nm, and the content of microalloying elements of the this steel is low, the number of MC carbides is also very small, so the carbide that really plays a role in performance is coarsened M3C , So we only mentioned the observation here, but did not focus on the discussion of MC.
Please put this answer in the paper. Now I know why you don’t include the MC carbides in the analysis, but the readers won’t. At the beginning of this part you can put this answer as the introduction, which explain why you don’t take MC carbides in account.
Author Response
Dear Reviewers:
Thank you for your letter and for the reviewers’ comments concerning our manuscript entitled “Effects of high-temperature tempering on mechanical properties and microstructure of SA738 Gr.B steel” (ID:metals-905981 ). The main corrections in the paper and the responds to the reviewer’s comments are as flowing:
1. Response to comment: JMatPro calculation
Response:we used the general steel module to calculate and we have added it into the paper(line 71)
2. Response to comment: MC carbides
Response: we have added the sentence Because of the low content and small size of MC carbides, the precipitation strengthening and the damage to toughness are limited, so we can ignore the discussion of MC carbides in this paper in the line 127-129,after we haved observed the MC by TEM.
Special thanks to you for your good comments.